# A New Financing Model for Tuberculosis (TB) Care in China: Challenges of Policy Development and Lessons Learned from the Implementation

**DOI:** 10.3390/ijerph17041400

**Published:** 2020-02-21

**Authors:** Qian Long, Weixi Jiang, Di Dong, Jiaying Chen, Li Xiang, Qiang Li, Fei Huang, Henry Lucas, Shenglan Tang

**Affiliations:** 1Global Health Research Center, Duke Kunshan University, Suzhou 215316, Jiangsu, China; weixi.jiang@dukekunshan.edu.cn (W.J.); di.dong@dukekunshan.edu.cn (D.D.); shenglan.tang@duke.edu (S.T.); 2School of Policy & Management, Nanjing Medical University, Nanjing 211166, Jiangsu, China; jychen@njmu.edu.cn; 3Tongji Medical College of Huazhong University of Science and Technology, Wuhan 430074, Hubei, China; xllyf@126.com; 4School of Public Health, Xi’an Jiaotong University, Xi’an 710049, Shanxi, China; tjlq@mail.xjtu.edu.cn; 5National Center for Tuberculosis Control and Prevention, China CDC, Beijing 102206, China; huangfei@chinacdc.cn; 6Institute of Development Studies, University of Sussex, Brighton BN19RE, UK; h.lucas@emeritus.ids.ac.uk; 7Duke Global Health Institute, Duke University, Durham, NC 27710, USA

**Keywords:** health financing, TB care, health policy and system research, China

## Abstract

*Background*: With support from the Gates Foundation, the Chinese Center for Disease Control and Prevention (China CDC) introduced a new financing model for tuberculosis (TB) care. This paper reviews the development of the associated financing policies and payment methods in three project sites and analyzes the factors impacting on policy implementation and outcomes. *Methods*: We reviewed policy papers and other relevant documents issued in the project sites. Semi-structured qualitative interviews were conducted with key stakeholders at provincial, city and county levels. Thematic analysis was applied to identify themes and develop interpretations. *Results*: The China CDC guideline proposed the introduction of a case-based payment based on TB treatment clinical pathways, increased reimbursement rates and financial assistance for the poorest TB patients. Contrary to expectations, TB patients with complications and/or comorbidities were often excluded from the program by hospitals that were concerned the cost of care would exceed the case-based payment ceiling. In addition, doctors frequently prescribed services and/or drugs beyond the coverage of the benefit package for those in the program. Consequently, actual reimbursement rates were low and poor patients still faced a heavy financial burden, though the utilization of services increased, especially by poorer patients. Qualitative interviews revealed three main factors affecting payment policy implementation. They were: hospital managers’ concern on the potential for reduced revenue generation; their fear that patients would regard the service provided as sub-standard if they were not prescribed the full range of available treatments; and a lack of mechanisms to effectively monitor and support the implementation process. *Conclusions*: While the intervention had some success in improving access to TB care, the challenges of implementing the policy in what proved to be an unreceptive and often antagonistic context resulted in divergences from the original design that frustrated its aim of reducing the financial burden on patients.

## 1. Introduction

The World Health Organization (WHO) End tuberculosis (TB) strategy (2016–2035) has laid out milestones and targets. These include: by 2030 a 90% reduction in the number of TB deaths and an 80% reduction in TB incidence compared with 2015; and by 2020 no TB-affected household facing catastrophic payment for TB care [1]. The first WHO Global Ministerial Conference on Ending TB, in 2017, and the first UN General Assembly high-level meeting on the fight against TB, in 2018, have driven international political action [2,3,4]. It has been acknowledged that profound health systems challenges, for instance in terms of financing TB care, will have to be overcome to meet the TB milestones and targets.

There has been a significant decline in the prevalence rate of smear-positive pulmonary TB in China over the two and a half decades since the WHO-recommended Directly Observed Treatment, Short-course (DOTS) strategy was implemented in the public health system, a reform led by the Chinese Center for Disease Control and Prevention (CDC) [5]. However, a recent WHO report estimates that in 2017 China still had the second highest TB incidence, with some 889,000 new cases [6]. Furthermore, the TB prevalence rate in the less developed western region was almost three times that in the eastern [7], reflecting a huge disparity in access to TB care across areas with different levels of economic development. The international literature indicates that even when, as in China, there is a policy of providing free essential TB care, the financial burden on TB patients in many developing countries is one of the main obstacles to timely diagnosis and completion of treatment [8].

With support from the Gates Foundation, the China CDC developed a series of interventions, based on studies undertaken over 2014–2015, to tackle TB-related challenges. One of the most important involved the design and implementation of new financial model of TB care. These were piloted in three prefecture-level cities (Zhenjiang, Yichang, and Hanzhong), located in three provinces at different levels of economic development (Jiangsu, Hubei, and Shaanxi). By working with three health insurance schemes: the urban employee basic medical insurance (UEBMI), the urban resident basic medical insurance (URBMI), and the rural new cooperative medical scheme (NCMS), the aim was to provide insured TB patients with improved benefit packages to reduce the financial burden of TB care.

Successful policy implementation depends heavily on an understanding of context and this is particularly important in China where local institutions (e.g., governments, social security agencies, and health facilities) are granted a considerable degree of autonomy and detailed negotiation is the primary mechanism for policy change [9]. The China CDC led and coordinated such negotiations across the relevant institutions in each study site. A consortium of academic institutions led by Duke University was commissioned to evaluate this intervention. This paper reviews the development of financing policies and payment methods for TB care in the three project sites and analyzes the factors impacting on policy implementation and outcomes in terms of the TB services provided, cost containment and financial burden on patients. The findings are intended to have significant implications for financial policy development and implementation.

## 2. Methods

### 2.1. Contexts

As indicated in Table 1, the three prefectural cities differ substantially in terms of economic development, income levels and TB incidence (Jiangsu province, Hubei province, and Shaanxi province statistical year book 2015 and health statistical year book 2015). Zhenjiang is a major center of industry in Jiangsu, the province with the second-highest level of Gross Domestic Product (GDP) in China. Yichang is a major transit port in Hubei province. Hanzhong is in a mountainous region of the north-western Shaanxi province, where fossil fuel extraction is the primary industry. Though starting from very different levels, all three sites are experiencing rapid GDP growth. There are substantial variations in income levels for the urban and rural populations across the cities. In less developed Yichang and Hanzhong, the reported TB incidence rates were higher than that in relatively developed Zhenjiang.

Since the 1990s, TB dispensaries or TB departments in prefectural and county/district CDC centers were in charge of the diagnosis of TB suspects, their treatment and management applying the DOTS strategy, and referral of patients with complications to public hospitals [5]. The recent national TB control plan proposed integration of TB care into specialized infectious diseases hospitals, or general hospitals with TB departments, to provide diagnosis and treatment and to cooperate with primary health facilities and local CDC centers on case management [10]. This so called ‘designated hospital model’ was introduced in 2002 in Zhenjiang, 2011 in Yichang and 2012 in Hanzhong. According to the national TB practical guideline, sputum smear test, sputum culture and X-ray examination are often recommended for TB diagnosis. Vast majorities of TB patients only require outpatient care. The treatment often lasts for six or eight months.

Financing for TB care often includes the earmarked fund from the central government, local government compensation, health insurance coverage and co-payment. The national TB control program provides the first-line anti-TB drugs and essential TB tests (e.g., sputum smear test and X-ray examination) for patients for free. However, other ancillary drugs and examinations (e.g., liver protection drugs, liver or kidney function examinations, etc.) are charged and often costly. By 2012, the three prefectural cities had almost universal health insurance coverage. However, there was no, or very limited, reimbursement for TB outpatient care by the three main health insurance schemes. At county hospital level, the reimbursement rates for TB inpatient care ranged from 60% to 80% under the different NCMS and URBMI schemes in the three cities, and reached 90% for those enrolled in the UEBMI after payment of a deductible. Overall, the reimbursement rate for inpatient care was lower at higher level hospitals [11]. TB designated hospitals largely rely on fee-for-services payment.

### 2.2. Data Sources

This study is set within the framework of health policy and systems research [12]. Our aim was to compare the original policy guidelines to the policies adopted in each study site and then to explore how those policies were implemented in the context of the existing health systems. Mixed data sources and methods were used. According to the personal communications and stakeholders’ interviews, we identified and collected policy papers and other relevant documents issued by the China CDC and other institutions in the three project provinces and prefectural cities. We developed a data extraction form to map out the China CDC suggested financing and payment policy for TB care and the policies issued in the project sites, which included beneficiaries, services package, deductible, reimbursement proportion, proportion of out-of-pocket payment, and payment methods by three basic health insurance schemes respectively as well as eligibility and coverage by medical financial assistance across different project sites. We also conducted a series of semi-structured qualitative interviews including key informant interviews with the main stakeholders at provincial, city and county levels (e.g., health policy-makers and administrators, CDC officials, hospital managers, health insurance agency staff, and Medical Financial Assistance agency staff), and focus group discussions with TB health professionals at TB designated hospitals (including TB doctors, public health nurse, lab technician, and hospital staff working at health insurance office in the hospital) to understand the policy development and implementation process and explore factors impacting on policy translation. The project evaluation team identified qualitative participants, and the project management office coordinated and invited the stakeholders for interview. Senior researchers from the project evaluation team undertook individual in-depth interviews and focus group discussion in a private room after obtaining interviewees’ consents. One junior researcher made the notes and recorded the interviews. All data were transcribed into the words. Thematic analysis was applied to identify themes and develop interpretations. 

### 2.3. Ethical Considerations

The project was reviewed and approved by the Institutional Review Board of Chinese Center for Disease Control and Prevention. All interviews were conducted with the informed consent of participants. The audio recordings and transcriptions were stored under password protection and only accessible by the evaluation team members. Data acquired by qualitative interviews will be stored for five years after the project is over and then destroyed.

### 2.4. Study Limitations

The development of a new financing policy for TB care allows adaption to local circumstances. This tends to make a case-study approach the only viable alternative. During the project period (2012–2015), the national health system reform encouraged to remove mark-ups on drug prices and integrate URBMI and NCMS schemes. In this study, we did not explore the impact of the ongoing reform on acceptance of a new financing model for TB care in the three project sites.

## 3. Results

### 3.1. Financing and Payment Model Design

The most important intervention introduced by the China-Gates TB project Phase II was the design and implementation of an innovative financing and payment model aiming to standardize TB treatment and reduce the financial burden placed on TB patients. The China CDC issued a practical guide to the financial policy design, which defined the target population, content of care packages based on TB treatment clinical pathways, sources of finance, case-based payment mechanisms, and a payment ceiling for a full course of treatment (Box 1). It proposed that health insurance should cover 70%–80% of the cost for a full course of TB treatment, with patients paying no more than 20%–30% out-of-pocket. The guide further suggested that the Medical Financial Assistance agency should provide financial support to cover the out-of-pocket payment for the poorest TB patients or those facing severe financial difficulties.

Box 1China Center for Disease Control and Prevention (CDC) guidelines for the innovative financing and payment policy for tuberculosis care.**China-Gates tuberculosis (TB) project Phase II**: pilot for a new financing and payment policy for TB care undertaken in Zhenjiang, Yichang and Hanzhong, aiming to standardize TB treatment and reduce the financial burden placed on TB patients.**Targets**: Active TB patients who are enrolled in local New Cooperative Medical Scheme (NCMS), Urban Residence Basic Medical Insurance (URBMI) or Urban Employee Basic Medical Insurance (UEBMI) and undertaking TB treatment at local TB designated health facilities. In addition to health insurance coverage, those who are from very low income households or face severe financial difficulties or are aged 65 and above, are eligible for medical financial assistance. **Contents of TB care**: Services packages for outpatient and inpatient TB care were developed. For outpatient care, the services package recommends routine blood, urine, liver and kidney tests once a month during the treatment, 3–4 sputum smear tests and 2 chest X-ray examinations or 1 chest CT as necessary (≤10% of patients). For inpatient care, the services package recommends hospitalization for no longer than 21 days and includes intramuscular and intravenous injections and other routine hospital inpatient services. In addition, the package also recommends therapies for TB patients with side effects or complications.**Financing and payment method for TB care**: Based on the defined service packages, multiple institutions, including the district/county health bureau, health insurance agency, Medical Financial Assistance agency, CDC and TB designated health facilities will estimate and reach consensus on the payment for a full course for TB treatment. A case-based payment method will be applied. Health insurance agencies are required to contribute 80% of the payment in Zhenjiang and Yichang and 70% in Hanzhong. TB patients will pay no more than 20% or 30% of the cost. For the poorest TB patients who are eligible, the Medical Financial Assistance agency should provide the patient contribution.Data source: China CDC official letter, 2013.

The China CDC organized the consultation meetings with key stakeholders from TB designated hospitals, health insurance agencies, and the Medical Financial Assistance offices in the three prefectural cities to explain the proposed financial policy and drive the related policy development. However, the proposed design was not favored by TB designated hospitals and health insurance agencies. On the one hand, the managers of TB designated hospitals argued that the TB clinical pathways were not suitable for all TB patients, particularly those with comorbidities and severe complications who may require intensive care far beyond the defined services packages. On the other hand, the revenue generated from TB care accounted for around one third of total hospital revenue at city designated TB hospitals in the three prefectural cities and managers were worried that there would be a large shortfall in revenue generation after the implementation of case-based payments. In addition, officials from the health insurance agencies were concerned that they would have to pay considerably more in total to TB designated hospitals given that most TB patients had only outpatient care, which had not previously been reimbursed. The negotiations lasted for almost half a year before a consensus was reached and the policies implemented.

Compared to the original guidelines that proposed a case-based payment ceiling for a full course of treatment (including outpatient and inpatient care), a significant variation in the agreed payment mechanism involved the definition of separate payment packages for outpatient and inpatient care, alongside a requirement that hospital admission rates should be at most 30%. In the three prefectural cities, active TB patients who were enrolled in the local NCMS, URBMI, or UEBMI and undergoing TB treatment at a local TB-designated health facilities were to be included under the new policy, with the health insurance agencies increasing their reimbursement rates (Table 2). In Zhenjiang, the estimated cost of outpatient care was 3000 Renminbi (RMB; equivalent to 427.50 United States Dollar at time of publication) and of inpatient care 8000 RMB. TB patients would pay no more than 20% out of pocket, i.e., a maximum 600 RMB for outpatient care and 1600 RMB for inpatient care. In Yichang, the payment method was similar to Zhenjiang, but costs were separately estimated by city and county level hospitals. In Hanzhong, new and retreated patients who only had outpatient care would pay no more than 900 RMB and 1200 RMB, respectively. If they had both outpatient and inpatient care, the estimated cost was 9000 RMB and patients would pay no more than 2700 RMB. In addition, the Medical Financial Assistance agency agreed to cover 60%–70% of out of pocket payments for the poorest TB patients in Yichang and Hanzhong, and provided transportation and nutrition subsidies for these patients in all three prefectural cities.

### 3.2. Implementation of New Financing and Payment Model and Factors Affecting Impacts

The agreed financing and payment policies were substantially weakened during the implementation process. First, many active TB patients who were expected to be included were in practice excluded. In Yichang and Hanzhong, this applied to many patients with complications and/or comorbidities, mainly due to a perception that additional services would be needed at a higher cost than the ceiling specified by the defined package. Second, actual reimbursement rates for patients was lower than suggested. In Zhenjiang, all eligible TB patients were included, but the effective reimbursement rates for both outpatient and inpatient care were substantially lower than the suggested 80%. Services within the defined package were reimbursed according to the new policy, but the cost of additional prescribed drugs (e.g., drugs for liver protection, etc.) and/or examinations beyond those specified were reimbursed based on the old reimbursement rates. Similarly, in Yichang and Hanzhong, if the cost for patients treated under the new policy was higher than the ceiling, the excess was reimbursed according to the previous reimbursement policy for both outpatient and inpatient care. Third, service provision was often far beyond that defined in the service packages described above. Moreover, in all three prefectural cities, hospital admission rates were higher than the recommended 30%. Finally, while the medical financial assistance policy was implemented as proposed, qualitative interviews with officials from the local CDC and Medical Financial Assistance agency reflected that though many TB patients were poor, only very few were eligible and benefited from the additional support. Consequently, many officials from the health authorities, CDC, and health insurance agencies argued that the cost of TB treatment was not under control as expected, and some of them said that poor patients still faced a heavy financial burden, although the utilization of services increased by TB patients, especially poor patients.

Qualitative interviews with key stakeholders revealed three main factors impacting on policy implementation. The potential for insufficient hospital compensation was most frequently mentioned. The great majority of hospitals rely on fee-for-services payments. Moreover, the salaries of health care providers are directly linked to revenue generation. In the three project sites, provincial and city level health authorities and CDC officials were concerned that implementation of the new payment policy could lead to a reduction in hospital revenue from TB care. Clearly, this would cause particular difficulties for policy implementation in city TB designated hospitals, where TB treatments accounted for a high proportion of total hospital revenue. They also said that hospital admission rates were “out of control” (ranging from 40% to 70%) and one of the main reasons was to increase profitability by providing more and more expensive services. Some health authority and CDC officials thought hospital managers and TB care providers would not implement the new payment policy unless they received additional compensation.

“The difference of income is obvious (hospital revenue before and after implementing new payment policy). Thus, government compensation should be given to ensure (health care providers’) salaries and hospital profits…otherwise, they (TB designated hospitals) would earn in other ways. For example, many (TB) patients were not included in the program. Although these patients were eligible, they (hospitals) defined they were not eligible and treated them in the original way (based on fee-for-services payment)”. (Jiangsu provincial CDC official, in-depth interview)

“The hospital admission rate is not regulated well in the city (TB designated) hospital. (In the study period), the lowest rate was 57%. Sometimes, he (doctor) purposely diagnosed ordinary TB (who do not require inpatient care) as TB with complications and then those are reasonably to be admitted”. (Yichang city CDC official, in-depth interview)

“Doctors’ salary is paid according to the old way (linked with revenue generation). So if you want him to control the hospital admission rate or reduce prescriptions for examinations or drugs, it is very difficult”. (Hanzhong city CDC official, in-depth interview)

Tensions between patients and health care providers was another concern. A number of health authorities and CDC officials thought adherence to the TB treatment clinical pathway had improved to some extent. However, most hospital managers and TB care providers in all three prefectural cities thought the tests and drugs covered by the package were too restrictive. Doctors were fearful of patients complaining that they were not getting the best treatment and thus often prescribed more and more expensive examinations, for example using a CT scan for diagnosis rather than X-ray, or admitting patients for reasons other than those specified by the defined clinical pathway. One hospital manager in Yichang reported that “the requirement for hospital admission is not consistent as between the health insurance agency and the China-Gates Phase II program. The insurance agency allows more latitude (for hospital admission). Doctors will prioritise safety of medical practice and admit patients when possible.”

There also was a lack of mechanisms to supervise and support implementation of the new policies. In all three prefectural cities, the local health administration authorized the CDC to coordinate between TB designated hospitals, health insurance agencies and the Medical Financial Assistance agency to monitor the implementation process. However, most CDC officials said that they found it very challenging to supervise TB treatment and regulate hospital behavior due to their limited clinical expertise and the lack of effective administrative sanctions. Many officials of health insurance agencies argued that it was hard to standardize the implementation of the case-based payment mechanism and control costs given the information asymmetry between the agency and the hospital and insufficient human resources to supervise the behavior of the hospital. For example, some reported that the primary diagnosis was sometimes changed from TB to another disease if the treatment expenditure was likely to exceed the ceiling for TB case-based payment. Often the explanation from the hospital was ‘uncertainty during the treatment’. Thus, the insurance agency had to pay for both TB treatment within the case-based payment package and expenditures relating to the primary diagnosis according to the local policy on outpatient and inpatient care.

“When the (TB treatment) expenditure is low, they (TB designated hospital) would not say anything, because we (health insurance agency) pay the hospital according to the case-based payment. But when the expenditure is high and may exceed the ceiling, they often change (to other disease diagnosis). This is a serious problem. …To be honest, we only look at invoices reported by the hospitals. We have very limited human resource to conduct field assessment and supervision in hospitals. We cannot manage it”. (Hanzhong, one county director of health insurance agency, in-depth interview)

“The cost for outpatient care are almost under control…but the costs of many inpatient cases exceed the expenditure ceiling. They (TB designated hospital) said it (services are covered by the package) is not enough, and we (health insurance agencies) have no choice at all”. (Yichang, one county director of NCMS agency, in-depth interview)

## 4. Discussion

Many low- and middle-income countries (including China) aim to provide TB diagnosis and treatment free of charge, however, direct and indirect costs due to TB continue accounting for a high proportion of annual household income of TB patients [8,13,14]. In the past, China has received substantial international assistance in the fight against TB, but now it is essential to mobilize increased domestic resources to improve the accessibility and affordability of high quality TB care. The China-Gates project Phase II introduced a new model of financing by multiple institutions and case-based payment mechanisms based on TB treatment clinical pathways, aiming to standardize treatment and reduce the financial burden on patients. While this innovative intervention is to be welcomed, we have identified serious problems that have arisen in the translation of the policy to real life settings. In particular, we would argue that it is not yet robust enough to meet the challenges posed by a context in which hospitals are predominantly dependent on revenue generation from service fees and the institutions which might influence their behavior have limited capacity or incentives to do so.

Since the implementation of the cost recovery policy in the 1980s, public hospitals largely rely on fee-for-services revenue and are driven to maximize profits [15,16]. The prices for high-tech diagnostic services and treatments were set above cost and substantial mark-ups on drug prices were allowed, creating perverse incentives for hospitals to purchase expensive equipment and imported drugs [17,18]. Consequently, over-diagnosis, over-treatment and over-prescription became common, leading to health expenditure escalation and increased dissatisfaction among patients [19,20,21,22,23]. Uncontrolled profit-maximizing behavior and excessive growth of health expenditure cannot be reversed without effective oversight of public health facilities.

TB diagnosis and treatment is not an exception. Many TB patients require only outpatient care, and the Chinese government provides free first-line anti-TB drugs and X-ray and sputum tests for TB diagnosis and treatment. However, in this study we found excessive hospital admission rates and the majority of doctors prescribing services and drugs beyond those required for essential TB care as covered by the benefit package. The average out-of-pocket payment for TB treatment was higher after the implementation of the new policy, and half of TB patients faced catastrophic health expenditure [24], though this was partly a reflection of the increased utilization of services, especially by poorer patients [25]. Implementation was hindered by a combination of hospital profit-maximizing behavior, information asymmetry between the hospitals and insurance agencies, which limited the opportunities for systematic monitoring, and a lack of the meaningful sanctions required for effective supervision. In an insurance-based system, the task of influencing provider behavior can potentially be undertaken most effectively by the insurance agencies, given that they are the major source of revenue. However, as described above, those agencies were very reluctant to assume this task in relation to the new TB policy, indicating that they lacked both the capacity and incentives to do so.

The latest round of health system reform in China was launched in 2009 and has emphasized the role of government in health investment and governance [26]. Studies reviewing the reforms have acknowledged significant progress in expanding basic health insurance coverage, promoting primary health care and establishing an essential medicines scheme [27,28,29]. The in-depth reforms in these areas are seen as largely on the right track. By contrast, it is argued that there is little meaningful evidence from the piloting of public hospital reforms to guide relevant policy development. Yip et al. conclude that hospitals and practitioners have become powerful stakeholders that can block reforms even under the new forms of health governance [27]. What we learned from the policy development and implementation of new financing of TB care in three project sites indicated that a mechanism that encourage stakeholders from multiple institutions to take active roles in ensuring quality of care and cost containment is urgently needed.

## 5. Conclusions

The new financing model for TB care that was a central component of the China-Gates TB project Phase II had the potential to improve access to high quality TB care and reduce the financial burden on patients. However, as with many such interventions, the challenges of implementing the policy in what proved to be an unreceptive and often antagonistic context resulted in a substantial divergence from the original design that significantly reduced its impact.

## Figures and Tables

**Table 1 ijerph-17-01400-t001:** Population, economic development, incomes, and tuberculosis burden of the three cities (2014).

Project Sites	Population (Million)	GDP per Capita (RMB *)	Provincial GDP per Capita Growth Rate	Urban per Capita Disposable Income (RMB *) ^1^	Rural per Capita Net Income (RMB *) ^2^	Reported TB Incidence Rate/100,000
Zhenjiang	3.17	102,651	9.3%	35,752	17,617	42.6
Yichang	4.11	76,369	9.7%	25,025	11,837	82.6
Hanzhong	3.43	28,908	10.6%	24,605	7933	60.2

Data sources: provincial statistical yearbook, health statistical yearbook, and government websites. TB: tuberculosis; GDP: Gross Domestic Product. * 1 Renminbi (RMB) = 0.161 US Dollar, 2014. ^1^ Urban per capita disposable income= (total household income-tax-social security fee)/number of persons in the household. ^2^ Rural per capital net income is the rural counterpart of urban per capital disposable income. Basically, it equals to the total income of a rural household minus operational fees, tax, depreciation, and income transfer out of the household.

**Table 2 ijerph-17-01400-t002:** Financing and payment policies for tuberculosis (TB) treatment in designated hospitals in three prefectural cities.

Study Sites	Basic Health Insurance Schemes (BHIS)					Medical Financial Assistance (MFA)	
	Targets	Payment Methods		Proportion of Reimbursement		Eligibility	Financial Assistance
		Outpatient	Inpatient	Outpatient	Inpatient		
Zhenjiang	Active TB as the first diagnosis	Full course—3000 RMB	Inpatient cost: 8000 RMB	No less than 80%	No less than 80%	Rural/urban poverty households, those with serious disability, and other special cases	N/A
Yichang	Active TB as the first diagnosis *	City hospital: 4900 RMB County hospital: 3900 RMB	City hospital: 9000 RMB County hospital: 5000 RMB	80%	Smear negative: 80% Smear positive: 90%	Rural/urban poverty households, those with serious disability, and other special cases	For rural/urban poverty households: 70% of out-of-pocket payment Others: 60% of out-of-pocket payment
Hanzhong	Active TB	Full course- New patients: 3000 RMB; Relapsed patients: 4000 RMB	Full course—9000 RMB (including outpatient care)	70%	70%	(1) ≥65 elderly TB patients (2) Rural/urban poverty households, those with serious disability, and other special cases	Outpatients: 600 RMB subsidy for new patients and 800 RMB for relapsed patients. Inpatients: 1800 RMB subsidy

Data sources: Official papers jointly issued by health bureau, human resource and health insurance agency and bureau of finance in Zhenjiang, Yichang and Hanzhong. * Retreated smear negative TB patients were excluded from the program.

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
