# Peer review of "A New Financing Model for Tuberculosis (TB) Care in China: Challenges of Policy Development and Lessons Learned from the Implementation"

_ijerph, 2020, doi:10.3390/ijerph17041400_

Round 1

Reviewer 1 Report

The title didn't fully cover the content of the article. It isn't an innovative way of financing, but for China a new way. It's about TB care, and not about TB control. The policy isn't limited to a change in tb financing, but it was also a change in treatment schemes. It's important because in the implementation process the doctors aren't willing to accept the new schemes and try to maximise profits.

Introduction: you have to make clear in what extend the new approach was different in comparison to the situation before. How TB care was financed before and what are the changes in treatment schemes?

58-59. You described that China has a policy of free essential TB care. But in the rest of the text you described in many places the contribution of the patient and or his insurance company. So, it isn't free at all! There was already a big gap between the policy and reality?

Context. Can you be more explicitly describe how normally the tb diagnosis and care is organized and what extend the organization of the care in the three town is different to the national policy?

You nicely described the difference between the CDC policy and the factual implementation.

Nowhere you are describing the role of the national health authority in relationship to the tb control. If tb is a public health issue, why is it so difficult to regulate the control mechanisms? And to enforce treatment schemes?

Nowhere you are citing the way other countries does organise tb control programmes and their financing mechanisms.

There is a lot of literature about financing mechanisms and the promotion of the quality of care. It's seems a big problem in China but why didn't you compare your situation to other countries? In a scientific article this is really missing in the discussion.

Author Response

We thank the reviewers for valuable comments, which have prompted a number of modifications to the manuscript. Our responses are as follows:

Reviewer #1:

Comment

Authors´ response

1. The title didn't fully cover the content of the article. It isn't an innovative way of financing, but for China a new way. It's about TB care, and not about TB control. The policy isn't limited to a change in tb financing, but it was also a change in treatment schemes. It's important because in the implementation process the doctors aren't willing to accept the new schemes and try to maximise profits.

Thank you for the suggestion. In this study, a case-based payment based on TB treatment clinical pathways was introduced aiming to control cost and ensure the quality of care. Thus, we modified the title: “A new financing model for Tuberculosis (TB) care in China: Challenges of policy development and lessons learned from the implementation”.

2. Introduction: you have to make clear in what extend the new approach was different in comparison to the situation before. How TB care was financed before and what are the changes in treatment schemes?

We have added and described in the context section (page 3, line 108-118). 

3. 58-59. You described that China has a policy of free essential TB care. But in the rest of the text you described in many places the contribution of the patient and or his insurance company. So, it isn't free at all! There was already a big gap between the policy and reality?

We added explanation in the context section (page 3, line 108-118) and discussed this issue in the text (page 9, line 310-323).

4. Context. Can you be more explicitly describe how normally the tb diagnosis and care is organized and what extend the organization of the care in the three town is different to the national policy?

We added and described the organization of TB diagnosis and care according to the national TB control practical guideline (page 3, in line 98-107). Three project sites have completed the transition of TB control model according to the national policy.

5. You nicely described the difference between the CDC policy and the factual implementation.

Nowhere you are describing the role of the national health authority in relationship to the tb control. If tb is a public health issue, why is it so difficult to regulate the control mechanisms? And to enforce treatment schemes?

We added the information related to financing for TB care in page 3 (line 108-118) and discussed the health governance in page 9, line 310-334.

6. Nowhere you are citing the way other countries does organise tb control programmes and their financing mechanisms.

There is a lot of literature about financing mechanisms and the promotion of the quality of care. It's seems a big problem in China but why didn't you compare your situation to other countries? In a scientific article this is really missing in the discussion.

We added the discussion in page 8 line 289-291. Many low- and middle-income countries aim to provide TB diagnosis and treatment free of charge. However, a systematic review showed many patients still suffered from catastrophic payment caused by TB. The main cost drivers varied by country, and thus the financial protection strategies also varied. In China, the major financial protection mean is through health insurance coverage. This paper focuses on the analysis of factors impacting on the implementation of financial policy for TB care in the Chinese context rather than evaluation of the intervention effects.

Reviewer 2 Report

Comments

What is the motivation to focus on the Chinese setting? What are the relevant limitations of the analysis that are presented in the paper? The concluding section of the paper should state more practical policy implications that stem from the results.

Author Response

We thank the reviewers for valuable comments, which have prompted a number of modifications to the manuscript. Our responses are as follows:

Reviewer #2:

Comment

Authors´ response

What is the motivation to focus on the Chinese setting?

China had the second highest TB incidence in the world. Like many other low- and middle-income countries, China aim to provide TB diagnosis and treatment free of charge within the health system, however, previous studies found financial burden placed on TB patients remained high. With support from the Gates Foundation, the China CDC introduced a new financial model of TB care. This paper reviews the policy development process and analyses the factors impacting on policy implementation and outcomes. The lessons learned from this study will be useful to other countries facing similar challenges.  

What are the relevant limitations of the analysis that are presented in the paper?

We have added in page 4, line 150-155.

The concluding section of the paper should state more practical policy implications that stem from the results. 

We proposed the policy implications in page 9, line 331-334.

Round 2

Reviewer 1 Report

Thanks for your reaction. I agree with your conclusion. But it's a pity that you didn't take the occasion to reflect a bit more about the way China is financing tb-control measures. Because, if doctors will be payed on the base of performance, a real problem has to be faced: how to build control mecanismes for quality of care. Otherwise it wil remain profitable for doctors to prescribe to much drugs and to promote the most profitable examens. But probably it isn't unique for tb-control, but for the whole system. It's inherent to performance based financing on quantity without the control of quality of care.